# Preload Dependency of 2D Right Ventricle Speckle Tracking Echocardiography Parameters in Healthy Volunteers: A Prospective Pilot Study

**DOI:** 10.3390/jcm11010019

**Published:** 2021-12-21

**Authors:** Christophe Beyls, Yohann Bohbot, Matthieu Caboche, Pierre Huette, Guillaume Haye, Hervé Dupont, Yazine Mahjoub, Abou-Arab Osama

**Affiliations:** 1Department of Anaesthesiology and Critical Care Medicine, Amiens University Hospital, 80054 Amiens, France; caboche.matthieu@chu-amiens.fr (M.C.); huette.pierre@chu-amiens.fr (P.H.); haye.guillaume@chu-amiens.fr (G.H.); dupont.herve@chu-amiens.fr (H.D.); mahjoub.yazine@chu-amiens.fr (Y.M.); abouao@gmail.com (A.-A.O.); 2UR UPJV 7518 SSPC (Simplification of Care Complex Surgical Patients) Research Unit, Jules Verne University of Picardie, 80000 Amiens, France; 3Department of Cardiology, Amiens University Hospital, 80054 Amiens, France; yohann.bohbot@chu-amiens.fr

**Keywords:** right ventricle, speckle tracking, strain, preload, fluid challenge

## Abstract

(1) Background: Right ventricular (RV) strain parameters derived from the analysis of the tricuspid annular displacement (TAD) are emergent two-dimensional speckle tracking echocardiography (2D-STE) parameter used for the quantitative assessment of RV systolic function. Few data are available regarding 2D-STE parameters and their dependency on RV preload. Our aim was to evaluate the effect of an acute change in RV preload on 2D-STE parameters in healthy volunteers. (2) Methods: Acute modification of RV preload was performed by a fluid challenge (FC): an infusion of 500 mL of 0.9% sodium chloride was given over 5 min in supine position. Preload dependency (responder group) was confirmed by a stroke volume increase of at least 10% measured by echocardiography. (3) Results: Among 32 healthy volunteers, 19 (59%) subjects were classified as non-responders and 13 (41%) as responders. In the responder group, the tricuspid annular plane systolic excursion (TAPSE) significantly increased (20 (20–23.5) mm to 24 (20.5–26.5) mm; *p* = 0.018), while RV strain parameters significantly decreased after FC: −23.5 ((−22.3)–(−27.3))% to −25 ((−24)–(29.6))%; *p* = 0.03) for RV free wall longitudinal strain and −22.8 ((−20.4)–(−30.7))% to −23.7 ((−21.2)–(−27))%; *p* = 0.02) for RV four-chamber longitudinal strain. 2D-STE parameters derived from the TAD analysis were not influenced by the FC (all *p* > 0.05). (4) Conclusions: In young, healthy volunteers, RV strain parameters and TAPSE are preload dependent, while TAD parameters were not. The loading conditions must be accounted for when evaluating RV systolic function by 2D-STE parameters.

## 1. Introduction

Two-dimensional (2D) transthoracic echocardiographic (TTE) assessment of the right ventricular (RV) systolic function is challenging, requiring a multi-parametric approach due to the complex geometry of the RV and its retrosternal location but also to the loading dependency of most echocardiographic parameters [1,2]. Conventional 2D echocardiographic assessment of the RV systolic function includes parameters evaluating only the longitudinal contraction of the RV lateral free wall, measured at the level of the tricuspid annulus, such as the tricuspid annular plane systolic excursion (TAPSE) and the systolic tricuspid annular velocity (RV-S’) [1], and also RV global systolic function parameters, such as the RV fractional area change (FAC). TAPSE and RV-S’ are easy to perform and reproducible, but their major limitation is their angle dependency [1].

Two-dimensional speckle-tracking echocardiography (2D-STE) is a relatively new, non-invasive echocardiography technique that allows an objective and quantitative assessment of RV systolic function [2] using simple parameters, such as RV longitudinal strain [3] and tricuspid annular displacement (TAD) [4]. The main advantages of 2D-STE parameters are their angle-independence [5] and their reproducibility [6] but also their established prognostic value (which is not the aim of the present experiment) [2,7]. TAD is an emerging 2D-STE parameter-tracking annular tissue toward the RV apex to assess the RV systolic function [8] by using three parameters, generated by a dedicated software.

TAD parameters are angle-independent, but their main advantage compared to RV longitudinal strain is their independence regarding endocardial borders definition [3,4,8].

However, current guidelines do not provide references values for 2D-STE parameters but rather lower limits of normality for vendor-dependent software [9]. Recently, international recommendations have been made to standardize the measurement, interpretation, and use of 2D-STE parameters [3].

Besides, recent studies have shown that under specific conditions, 2D-STE parameters are dependent on the patient’s blood volume [9,10] and therefore suggest that their interpretation should be made with caution [10]. There are no data regarding the direct influence of an acute change in preload induced by fluid challenge (FC) on 2D-STE parameters in patients without cardiopulmonary disease. Previous experimental studies used surrogates of FC, such as passive leg raising (that mobilizes blood from the venous compartment), to increase RV preload [11,12].

According to the Frank Starling relationship, we hypothesized that an FC-induced acute increase of RV volume should be followed by an increase in RV systolic function and therefore by an increase in RV 2D-STE parameters value.

The aim of this study was to evaluate the effect of an acute change in preload on 2D-STE parameters in healthy volunteers. This acute change in preload was performed by an infusion of 500 mL of saline over 5 min. Preload dependency of 2D-STE parameters was defined by a 10% increase of stroke volume (SV) measured by transthoracic echocardiography after FC.

## 2. Materials and Methods

### 2.1. Study Population

This study was an ancillary study of the PORTEAU study (NCT03589261) evaluating portal blood flow after FC at Amiens University Hospital between September 2018 to November 2019. We retrospectively analysed echocardiographic data from 37 healthy volunteers who participated in this study. We excluded 5 patients in whom it was difficult to assess RV speckle tracking analysis because of poor imaging quality. The final study population consisted of 32 healthy volunteers (Figure 1).

### 2.2. Definition of Preload Dependency

The acute modification of the subjects’ RV preload was performed by a FC [11]. According to the Frank–Starling relationship [12], preload dependency (responder group) was confirmed by an increase of at least 10% of the SV measured by echocardiography after the FC [10,11]. Comparative analyses were performed between the responder group and the non-responder group for each parameter to detect changes in 2D-STE parameters after FC.

### 2.3. Protocol Study

The volunteers’ blood volume was standardized as best as possible. All volunteers had been fasting from 00:00 a.m. and were included in the study between 8:00 and 11:00 a.m. After baseline TTE, a FC was realised by the infusion of 500 mL of 0.9% sodium chloride (0.9% NaCl, Clear Flex, Baxter, Guyancourt, France) over 5 min via a 16-gauge peripheral vein line in the left arm. The post-FC TTE was performed 15 min after the FC. Subjects were monitored for 2 h after the post-FC TTE to detect any FC-related adverse events. We collected the following demographic data for all subjects: weight, height, and body mass index (BMI). The following baseline and post-FC experimental data were also collected: systolic arterial pressure (SAP), diastolic arterial pressure (DAP), and mean arterial pressure (MAP) with an automated pneumatic device and heart rate (HR) using a continuous 3-line electrocardiogram.

### 2.4. Echocardiography

A standardized TTE protocol was performed according to recent guidelines [1] and using high-quality commercially available ultrasound system (CX 50, Philips Healthcare). The standardized TTE protocol was performed in supine position by a single experienced cardiologist (CB). Two-dimensional digitalized videos were stored and analyzed offline (Xcelera, Philips Healthcare). The velocity-time integral of the left ventricular tract outflow (LVOT_VTI_) was averaged on three pulse consecutive Doppler measurements in a 5-chamber view [1]. The chamber area was calculated from the left ventricular outflow tract diameter (LVOT_area_) in a long axis parasternal view. Doppler stroke volume (SV) was estimated by the product of LVOT_area_ and LVOT_VTI_ [1]. RV systolic function was assessed using a multiparametric approach according to current guidelines, including TAPSE, RV S′, and FAC [5]. The respiratory variation of the inferior vena cava diameters (IVC collapsibility index) was measured in spontaneous respiration and calculated as follows: (Dmax-Dmin)/Dmax where Dmax and Dmin were the maximal and minimal diameters of the IVC.

### 2.5. D-STE Analysis

2D-STE parameters were analysed offline using the commercial software package available on cart with Philips CX50 echocardiography machine (Automated Cardiac Motion Quantification, QLAB version 9.0, Philips Medical systems, Andover, MA, USA). 2D-STE parameters were analysed in one loop, and results were the average of 3 measured. All 2D-STE parameters were performed offline by two observers (CB and MC) blinded to fluid responsiveness status.

### 2.6. RV Strain Analysis

For RV strain analysis, the left-ventricle-specific strain software was used because RV-specific software was unavailable. The region of interest (ROI) was generated automatically and adjusted manually whenever the automated one was of poor quality. A full-wall approach was used for RV strain analysis: endocardial border of the RV strain was manually traced at end systole and automatically adjusted to include the entire myocardium. The frame rate ranged between 60 and 80 frames/second. Apical four-chamber views focused on the RV were acquired for the 2D RV free wall longitudinal strain (RVFWLS) analysis according to current guidelines [3] (Figure 2A). RVFWLS was calculated as the average of the three segments (the basal, mid-ventricular, and apical segments). For RV four-chamber strain (RV4CSL), 6 segments were analysed in a classic apical four-chambers view. (Figure 2B). Segments in which adequate tracking quality was not obtained despite manual adjustment were excluded from the analysis. Longitudinal strain was defined as the percentage of myocardial shortening relative to the original length and presented as a negative value, with a more negative value of strain reflecting better shortening [5].

### 2.7. TAD Analysis

TAD parameters measurement were also performed on the apical four-chamber view.

Three points were used for initialization on the first diastolic frame. These points were placed (1) at the insertion of the anterior tricuspid valve leaflet (RV free wall) and (2) of the septal leaflet into the tricuspid annulus and (3) at the RV apex (Figure 2C). The software automatically tracked and calculated the TAD at the RV free wall (TAD_lat_) and at the interventricular septum (TAD_sep_). RV longitudinal shortening (RV-LS) was calculated as maximum end-systolic displacement (LES) of the mid-annular point from the measured annular motion and expressed in percent of the end diastolic RV longitudinal dimension (LED): 100 × (LED − LES)/LED (Figure 2C). The mid-annular point is automatically selected by the software. The echocardiographic method used to measure TAD parameters was identical to that described in previous studies [4,13].

### 2.8. Statistical Analysis

Data are expressed as mean ± standard deviation (SD), median (interquartile range) or numbers (percentage), as appropriate. Variables were compared using Wilcoxon–Mann–Whitney, chi-square, or Wilcoxon rank-sum tests, as appropriate. To evaluate the intra-observer and inter-observer variability for offline analysis, data of 10 healthy volunteers were randomly selected and analysed by two operators (CB and MC) with at least a 1-week interval between the two analyses. Each operator was blinded to the results of the other. The inter-observer and intra-observer’s reliability of 2D-STE measurements were assessed using intraclass correlation coefficient (ICC) as recommended [14].

All statistical analyses were performed with IBM SPSS software (SPSS, version 24, IBM, New York, NY, USA). The limit of statistical significance was *p* < 0.05.

## 3. Results

### 3.1. General Characteristic of the Study Population

Thirty-two healthy volunteers were included from September 2018 to November 2019. The median age was 26 (24–28) years, and 87% were male (*n* = 28). Hemodynamic parameters before and after FC are displayed in Table 1 and demonstrated reduction in SAP (119 mmHg (115–128) to 116 mmHg (110–121); *p* = 0.02) and in HR (64 bpm (58–71) to 60 bpm (54–69); *p* = 0.03) after FC. Left ventricular (LV) and RV echocardiographic parameters before and after FC are reported in Table 2. In the overall population, there was no significant difference after FC in terms of LV systolic or diastolic parameters, TAPSE, RV-S’, FAC, and RVFWSL. After FC, RV mid-cavity dimension increased (*p* = 0.01), while IVC collapsibility index decreased (*p* = 0.001). The only RV systolic parameter significantly affected by FC was RV4CSL (−22.6% (20.7–25.4) to −23.5 (22.0–26.1); *p* = 0.05).

### 3.2. Preload Dependency

After FC, 19 (59%) subjects were classified as non-responders and 13 (41%) as responders with an increase in SV of more than 10%. In the responder group, there was no significant difference for hemodynamic parameters before and after FC (all *p* ≥ 0.09) (Table 3). Regarding conventional RV parameters, TAPSE significantly increased after FC (20 (20–23.5) to 24 (20.5–26.5) mm; *p* < 0.018), while RV-S’ and FAC were not significantly improved (*p* = 0.44 and *p* = 0.31, respectively). RV strain parameters (RVFWLS and RV4CLS) were associated with preload dependency (−23.5% (22.3–27.3) to −25% (24–29.6); *p* < 0.03) and −22.8% (20.4–30.7) to −23.7% (21.2–27); *p* < 0.02, respectively) **(**Figure 3), while TAD parameters were not (all *p* ≥ 0.27). In the responder group, IVC collapsibility index decreased after FC (from 35.5 (28–55)% to 24 (22–30), *p* = 0.001).

### 3.3. Preload INDEPENDENCY

After FC, 19 (59%) patients did not increase their stroke volume more than 10% relative to baseline and were therefore in the non-responders group (Appendix A). In this group, 2D-RV conventional parameters, RV strain parameters, and TAD were not associated with significant changes after FC (all *p* ≥ 0.11).

### 3.4. Feasibility and Reproducibility

Overall, the feasibility of the speckle tracking parameters was good (86% (*n* = 32/37)). Appendix A showed that the reproducibility of RVFWLS, RVF4C, and TAD were excellent.

## 4. Discussion

To our knowledge, this is the first study to investigate preload dependency of RV 2D-STE parameters assessed by performing a FC of 500 mL of 0.9% sodium chloride in young healthy volunteers.

The main results of our study can be summarized as follows: (1) among 2D-STE parameters, RVFWLS, and RV4CSL are preload-dependent, while TAD seems preload-independent; (2) among 2D RV conventional parameters, only TAPSE is preload-dependent; and (3) the reproducibility of 2D-STE measurements is excellent.

Due to its large availability and safety, 2D echocardiography is the first-line non-invasive imaging modality used to assess the RV systolic function by a standardized multiparametric approach [5,9]. However, the location and the complex geometry of RV represent a major challenge for echocardiographic interpretation. Cardiac magnetic resonance (CMR) is currently the gold standard for RV systolic function assessment, but its availability is limited. [5].

The introduction of new echocardiographic parameters, such as 2D-STE parameters, has significantly improved our ability to assess RV systolic function. These parameters are highly correlated with CMR measurements even if CMR evaluate RV in three dimensions [13]. However, 2D-STE parameters are more available and feasible at the patient’s bed side. RV strain is an accurate, highly-feasible, fast, and reproducible echocardiographic method that provides important data about RV systolic function or RV mechanics [9]. In view of its increasing use in clinical practice, current guidelines have provided standardization of 2D-STE parameters for technical engineering and clinical community [3]. Furthermore, 2D-STE parameters are used to stratify the prognosis and to address the management of patients with a wide range of cardiovascular and pulmonary diseases [3,9]. Cardiovascular physiology has an impact on 2D-STE parameters. Experimental results and mathematical models have shown that strain deformation is influenced by loading conditions. Given this sensitivity to loading conditions, abnormal strain values are therefore not synonymous with myocardial disease, and normal values do not exclude disease [9]. Furthermore, RV function is more dependent on preload, contractility, and afterload than LV function. According to the Frank–Starling mechanism, the increase of myofiber stretching by an increase in preload results in a more forceful contraction and therefore increases RV strain (absolute) values [9,12].

### 4.1. Preaload in Healthy Subject

Cardiovascular adaptation to increased preload is a pathophysiological mechanism supported by many factors. According to Frank–Starling law, there is a positive relationship between preload and stroke volume. However, this relationship is not linear but rather traces a curve. Accordingly, once a concrete preload value has been reached, further increments do not give rise to significant additional stroke volume elevation. Preload dependency is the capacity of the heart to modify stroke volume in response to changes in preload and depends on the basal preload value and the zone of the Frank–Starling curve in which both ventricles are operating. Each individual patient can present a series of ventricle function curves and therefore a different response for the same preload increase. In hemodynamically unstable patients, the management of fluid therapy is crucial because only 50% of patients are preload responders [11]. In our study, 13 (41%) patients had an increase of stroke volume after the FC, but no clinical data were found to explain this behavior. Interindividual variations in body composition, basal circulating state, and coping with fasting may have influenced FC results. Future larger experimental studies are required to explore the different pathophysiological mechanisms involved in preload dependency of healthy volunteers

### 4.2. Preload Dependency of 2D-STE Parameters in Healthy Volunteers

To our knowledge, there are few data on RV preload dependency of 2D-STE parameters in healthy volunteers. Garcia-Lopez et al. [15] reported that strain analysis was reproducible (ICC 0.773) and sensitive to detect differences in RV4CLS after a passive leg-raising manoeuvre in 31 healthy volunteers. However, the strain analysis was not performed in the RV apical view but by with analysis of the six segments in a classical four-chamber view. Sano et al. [16] evaluated the RV contractile reserve using a preload stress echocardiography in patients with pulmonary hypertension compared to healthy volunteers. They reported that RV free wall strain (assess in the RV-focused apical four-chamber view) increased significantly (+3.5%) as well as SV (+8.4 mL) in the healthy volunteer’s group (15). Susilovic-Grabovac et al. [17] evaluated the effect of a single air self-contained underwater breathing apparatus (SCUBA) dive on RV function in 12 healthy professional divers. They showed a significant increase of RVFWLS, TAPSE, and right atrial volume in association with an increase of cardiac output 30 min after the dive, probably related to RV afterload reduction. These results suggested that RFWLS could reveal subclinical changes in RV function due to acute change of preload and afterload in healthy volunteers [17].

Our study showed a significant increase in RV4CLS (+2.4%) and in RVFWLS (+1.6%) after FC when SV increased significantly (+13 mL) (Figure 3), suggesting that RV strain can detect a subclinical increase in myocardial fibre stretching due to an acute increase of preload. It is important to point out that these changes in RV strain occurred in the absence of significant modifications in hemodynamic parameters. Indeed, changes in heart rate or blood pressure can modify RV strain values. Therefore, the increase in strain values was essentially due to an increase in myocardial contractility due to an acute increase in RV preload.

### 4.3. Preload Dependency of TAPSE and TAD in Healthy Volunteers

RV contraction is mainly related to the longitudinal contraction of fibres (up to 80%) that shorten the long axis and attract the tricuspid annulus toward the apex [18]. In order to assess the RV systolic function, several echocardiographic parameters, such as TAPSE or RV-S’, were developed. TAPSE is a single-dimensional, reproducible, and easily obtainable method for the assessment of RV function, which is routinely performed in clinical practice. It measures the maximal distance of systolic excursion of a single RV annular segment along its longitudinal plane [19]. TAPSE is an angle-dependent parameter, and guidelines suggest that TAPSE may be preload dependent [5]. Accordingly, we found that TAPSE increased after FC.

TAD is a fast, automated, and reproducible parameter that provides information on local displacement of the tricuspid annulus toward the apex. In our study, median TAD_lat_ was 22.7 (20.5–25) mm and 15.2 (13.7–17.3) mm for TAD_sep_. Those values are consistent with prior studies in healthy volunteers [4,8,19]. Indeed, in a healthy teenager population, mean values of TAD were 20.7 mm ± 3.7 for TAD_lat_ and 14 ± 1.9 for TAD_sep_ [20]. TAD lateral displacement was higher compared to the septal, mainly due to the interventricular coupling via the septum, resulting in anchoring of the septal side of the tricuspid annulus, which is less pronounced at the lateral side of the annulus [8].

To our knowledge, our study is the first to show that TAD parameters were preload-independent in healthy volunteers. Even if the measurements of TAPSE and TAD parameters are both based on the displacement of the tricuspid annulus, they are not interchangeable [19]. TAPSE measurement assesses excursion of the tricuspid annulus, while TAD_lat_ measurement evaluates its shortening toward the RV apex. Data on TAD parameters in situation of preload dependency are scarce. However, in clinical situation of increased afterload, TAD_lat_ parameters seemed to be superior to TAPSE for identifying RV dysfunction with a different cut-off value for each parameters [4]. Because afterload is intimately related to preload, it is likely that there may be differences in the measurement of these parameters during an acute increase in preload.

Due to their reproducibility, angle independence, and ease of measurement, TAD parameters can be used in many clinical situations and myocardial disease. TAD parameters can be use in operating room for monitoring RV dysfunction with transoesophageal echocardiography [21] or in daily practice for assessing RV dysfunction in pulmonary hypertension patients [4]. In a recent study, we demonstrated that TAD parameters, especially RV-LSF, had a diagnostic function of the RV dysfunction superior to 2D-STE RV strain or conventional RV parameters in acute respiratory distress syndrome due to COVID-19 infection in supine [21] or prone positioning [22].

### 4.4. Limitations

Our study has several limitations: first, it is possible that infusion of larger volume would have led to significant changes in 2D-STE parameters or in 2D conventional parameters. However, our results were comparable to those of the study using passive leg raising, which is a dynamic manoeuvre allowing to mobilize 300 mL of blood in order to induce a change in preload [11,15]. Moreover, different shapes, body compositions, and basal circulating states of the subjects may have influenced FC results. For further studies, a Bioelectrical impedance analysis, for example, may allow a more accurate assessment of basal state and fluid distribution [23].

Secondly, five volunteers were excluded for poor image quality. Indeed, the image quality and the RV focus view can impact the ability to measured 2D-STE parameters and thus affect the reproducibility of 2D-STE parameters. All echocardiograms were analyzed by a single vendor’s software, and the QLAB software had no specific ROI for RV strain analysis. One another important limitation is that, unlike RVFWLS or RV4CSL, which are widely available, TAD parameters are only provided by Philips Healthcare and not by other constructors. In our study, regional strain analysis was assessed in a 2D four-apical chamber view. Recent studies have shown that the three-dimensional approach can improve the assessment of global and regional RV systolic function [24,25]. However, this technique was not available in our study. Furthermore, it requires specific TTE probes and an ultrasound machine and is more time consuming.

The results of our study have not been compared with other techniques as CMR especially because CMR and echocardiography techniques are not interchangeable for RV strain evaluation [26]. Finally, this study involved a limited number of young volunteers, especially males, in a single centre. Further larger studies are required to confirm our findings.

## 5. Conclusions

Our study demonstrates that, in young healthy volunteers, RVFWLS, RV4CLS, and TAPSE are associated with preload dependency, while 2D-STE TAD parameters are not. Our study also shows that 2D STE parameters are highly reproducible. Therefore, this study shows that loading conditions should be taken into account to evaluate RV systolic function with 2D-STE parameters.

## Figures and Tables

**Figure 1 jcm-11-00019-f001:**
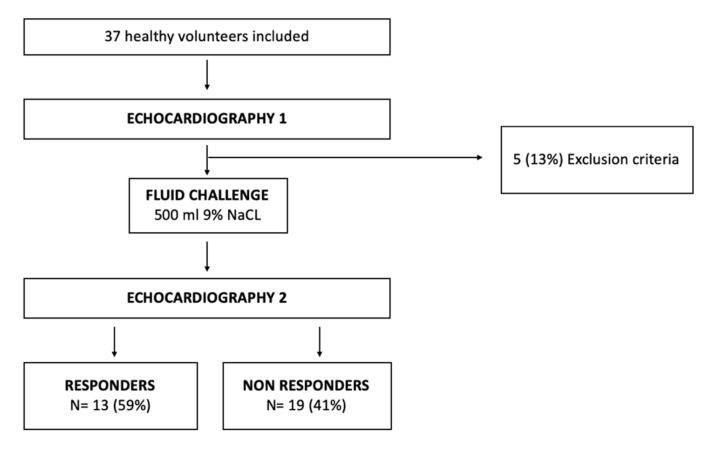
Flow chart of the study population.

**Figure 2 jcm-11-00019-f002:**
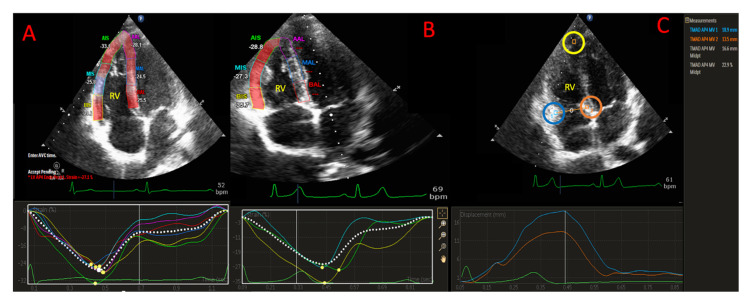
RV 2D-STE parameters. (**A**) Right ventricular four-chamber strain (RV4CLS), including the ventricular septum. (**B**) Right ventricular free wall longitudinal strain (RVFWLS) in focused right ventricular view. In focused right ventricular view, septal segments are not compatible with the septal segments from a standard LV segmentation, and results can therefore not be used interchangeably. (**C**) In RV-focused view, user-defined anatomic landmarks lateral point (blue circle) and septal point (orange circle) were placed at the bottom of the RV free wall and the bottom of the interventricular septum, and another point was placed at the apex (yellow circle). TAD lateral, TAD septal, and RV-LSF (%) value were displayed.

**Figure 3 jcm-11-00019-f003:**
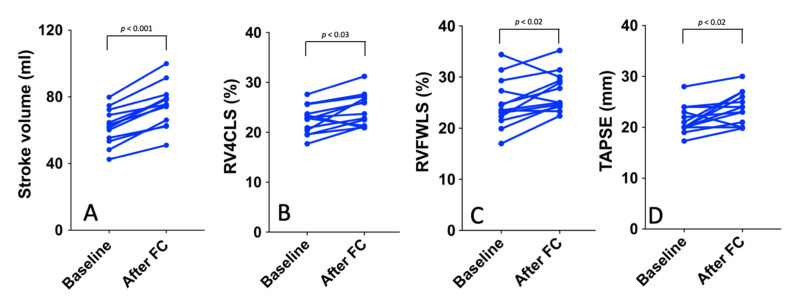
Stroke volume (**A**), RV4CLS (**B**), RVFWLS (**C**), and TAPSE (**D**) at baseline and after FC in the responder group (blue line).

**Table 1 jcm-11-00019-t001:** Haemodynamic parameters before and after fluid challenge.

Overall Population (*n* = 32)	Baseline	After FC	*p*-Value
Age (years)	26 (24–28)	-	-
BMI (Kg/cm^2^)	22 (20–24)	-	-
Male, *n* (%)	28 (87)	-	-
Haemodynamic parameters			
HR (bpm)	64 (58–71)	60 (54–69)	0.03
SAP (mmHg)	119 (115–128)	116 (110–121)	0.02
MAP (mmHg)	84 (76–90)	81 (75–89)	0.18
DAP (mmHg)	68 (64–77)	70 (62–75)	0.71

Continuous variables are expressed as median (interquartile range) and categorical variables as number (percentage). BMI, body mass index; DAP, diastolic blood pressure; FC, fluid challenge; MAP, mean blood pressure; HR, heart rate; SAP, systolic blood pressure.

**Table 2 jcm-11-00019-t002:** Echocardiographic parameters before and after fluid challenge.

Overall Population (*n* = 32)	Baseline	After FC	*p*-Value
LV systolic parameters			
LVEF (%)	61 (58–68)	60 (55–66)	0.18
LV end diastolic volume (mL)	113 (94–118)	111 (94–121)	0.29
LV end systolic volume (mL)	43 (34–49)	45 (34–52)	0.21
Stroke volume (mL s^−1^)	72 (61–79)	75 (65–80)	0.07
CO (L min^−1^)	4.6 (3.9–5.2)	4.2 (3.7–5.3)	0.52
LV diastolic functional parameters			
E wave (cm s^−1^)	84 (74–94)	88 (78–95)	0.09
A wave (cm s^−1^)	42 (38–54)	47 (40–59)	0.16
E/A ratio	1.8 (1.6–2.2)	1.8 (1.7–2.1)	0.93
Lateral E/e’	4.2 (3.8–5)	4.5 (4.1–5.4)	0.22
Deceleration time (ms)	204 (168–266)	202 (68–269)	0.7
LA volume index (mL m^−2^)	18 (20–23)	21 (19–23)	0.92
RV Parameters			
RVOT PSAX distal dimension (mm)	27 (24–32)	30 (27–33)	0.46
RVOT PLAX proximal dimension (mm)	31 (26–34)	31 (27–34)	0.11
RV basal dimension (mm)	40 (36–44)	39 (37–42)	0.78
RV mid-cavity dimension (mm)	3.6 (3.5–4)	3.8 (3.7–4)	0.01
RV longitudinal dimension (mm)	8.3 (8–8.5)	8.3 (7.9–8.5)	0.82
RV EDA (mL)	21 (16–23)	21 (18–34)	0.16
RV ESA (mL)	11 (9–15)	12 (10–15)	0.62
RA volume indexed to BSA (mL/m^2^)	23 (19–27)	24 (22–29)	0.63
IVC collapsibility index (%)	35.5 (28–55)	24 (22–30)	0.001
RV Systolic Function Parameters			
TAPSE (mm)	23 (20–27)	25 (21–28)	0.06
RV- S’ (cm/s^−1^)	15 (11–16)	14 (13–16)	0.9
RV FAC (%)	42 (33–48)	42 (37–46)	0.7
IVA (m s^−2^)	3.2 (2.5–3.7)	3 (2.5–3.5)	0.29
2D STE RV Strain			
RVFWLS (%)	−24.3 (22.5–28.9)	−24.9 (23.4–29)	0.34
RV4CLS (%)	−22.6 (20.7–25.4)	−23.5 (22.0–26.1)	0.05
TAD parameters			
❖ TAD_lat_ (mm) ❖ TAD_sep_ (mm) ❖ RV-LSF (%)	22.7 (20.5–25)	24.6 (21.3–26.3)	0.08
	15.2 (13.7–17.3)	16.1 (13.1–19.5)	0.44
	24.2 (22.2–27)	25 (21–29.4)	0.3

Continuous variables are expressed as median (interquartile range) and categorical variables as number (percentage). 2D-STE, bi-dimensional speckle tracking echocardiography; CO, cardiac output; IVC, inferior vena cava; LA, left atrial; LV, left ventricular; LVEF, left ventricular ejection fraction; PSAX, para-sternal short axis; PLAX, para-sternal longue axis; RA, right atrial; RV, right ventricle; RV EDA, Right ventricle end-diastolic area; RV ESA, right ventricle end-systolic area; RV-FAC, right ventricle fractional area change; RV4CLS, right ventricle four chamber longitudinal strain; RVFWLS, right ventricle free wall longitudinal strain; RV-LSF, right ventricle longitudinal shortening fraction; RVOT, right ventricular outflow tract; TAD, tricuspid annular displacement; TAPSE, tricuspid annular plane systolic excursion; VTI, velocity time integral.

**Table 3 jcm-11-00019-t003:** Hemodynamic and echocardiographic parameters in the responder group before and after fluid challenge.

Responder Group (*n* = 13)	Baseline	After FC	*p*-Value
Hemodynamic parameters			
HR (bpm)	64 (57–74)	61 (56–67)	0.09
SAP (mmHg)	119 (113–125)	112 (107–117)	0.25
MAP (mmHg)	79 (75–87)	77 (73–84)	0.41
DAP (mmHg)	65 (62–68)	67 (60–72)	0.93
LV systolic parameters			
LVEF (%)	59 (55–68)	59 (57–64)	0.65
LV end diastolic volume (mL)	112 (88–116)	113 (94–125)	0.24
LV end systolic volume (mL)	43 (32–51)	45 (34–52)	0.59
Aortic VTI (cm/s^−1^)	18 (16–20)	22 (19–25)	0.001
Stroke Volume (mL)	62 (54–70)	75 (64–80)	0.001
RV Function			
TAPSE (mm)	20 (20–23.5)	24 (21–29)	0.02
RV- S’ (cm s^−1^)	15 (11–16)	14 (13–16)	0.44
RV FAC (%)	42 (36–48)	43 (37–46)	0.31
IVA (m s^−2^)	3.1 (2.5–3.5)	3.0 (2.6–3.6)	0.44
RV Strain			
RVFWSL (%)	−23.5 (22.3–27.3)	−25.0 (24.0–29.6)	0.03
RV4CSL (%)	−22.8 (20.4–30.7)	−23.7 (21.2–27.0)	0.02
TMAD			
❖ TAD_lat_ (mm) ❖ TAD_sep_ (mm) ❖ RV-LSF (%)	22 (20–24)	24 (18–26)	0.27
	15 (14–18)	16 (13–20)	0.43
	24 (22–27)	25 (20–30)	0.63

2D-STE, bi dimensional speckle tracking echocardiography; CO, cardiac output; LA, left atrial; LV, left ventricular; LVEF, left ventricular ejection fraction; PSAX, para-sternal short axis; PLAX, para-sternal longue axis; RA, right atrial; RV, right ventricle; RV EDA, right ventricle end-diastolic area; RV ESA, right ventricle end-systolic area; RV4CLS, right ventricle four chamber longitudinal strain; RVFWLS, right ventricle free wall longitudinal strain; RV-FAC, right ventricle fractional area change; RV-LSF, right ventricle longitudinal shortening fraction; RVOT, right ventricular outflow tract; TAD, tricuspid annular displacement; TAPSE, tricuspid annular plane systolic excursion; VTI, velocity time integral.

## Data Availability

Data available on request due to restrictions, e.g., privacy or ethical.

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
