# Peer review of "Preload Dependency of 2D Right Ventricle Speckle Tracking Echocardiography Parameters in Healthy Volunteers: A Prospective Pilot Study"

_jcm, 2021, doi:10.3390/jcm11010019_

Round 1
Reviewer 1 Report
Dear authors,
I found your article very interesting, with a topic that is justified from a pathophysiological point of view, but less debated in the literature.
Thus I have some concerns:
In the methods, you have clearly stated that all measurements were performed by the same technician. Then what is the reason why you have also assessed the inter-observer reliability, see lines 167-170?
I have carefully read the manuscript and I found no explanation, or at least, supposition why some subjects are responders and some not. Also what could be the clinical significance of these types of responses?
In the limitation section, you could have added that these data were not compared with other methods, at least MRI, if not right cardiac catheterism (which is to understand because your subjects were healthy volunteers and the recordings were performed some time ago).
Please check again the grammar of the manuscript: there are several places where there are discrepancies between nouns and verbs.
Author Response
Reviewer 1
Dear authors,
I found your article very interesting, with a topic that is justified from a pathophysiological point of view, but less debated in the literature.
Thus I have some concerns:
In the methods, you have clearly stated that all measurements were performed by the same technician. Then what is the reason why you have also assessed the inter-observer reliability, see lines 167-170?
Response: We thank the reviewer for this remark. We apologize for this mistake. The acquisition of TTE images was performed by a single operator (CB). The off-line analysis of the images, for 2D-STE measurement, was performed by 2 operators (CB and MC) allowing to measure the inter and intra observer reliability. We modified and clarified the ultrasound method used page 4, line 182 as follow: “All 2D-STE parameters were performed off line by two observers (CB and MC) blinded to fluid responsiveness status. And page 4, line 164 as follow: The standardized TTE protocol was performed in supine position by a single experienced cardiologist (CB).
We completed the statistical part page 5 , line 186 as follow: To evaluate the intra-observer and inter-observer variability for offline analysis, data of 10 healthy volunteers were randomly selected and analyzed by two operators (CB and MC) with at least a 1-week interval between the two analyses. Each operator was blinded to the results of the other. The inter‐observer and intra‐observer's reliability of 2D-STE measurements were assessed using intraclass correlation coefficient (ICC) as recommended [16].
I have carefully read the manuscript and I found no explanation, or at least, supposition why some subjects are responders and some not. Also what could be the clinical significance of these types of responses?
Response: We thank the reviewer for this interesting remark. There is no data on preload dependence in healthy subjects after a fluid challenge. We know that in unstable patients, 50% of them had an increase of SV after a FC. But in our study no clinical data were found to explain the difference in preload dependence and future larger experimental studies are required to explore the different pathophysiological mechanisms.
We added a new section about preload dependency in the discussion section, page 12, line 378, as follow: Cardiovascular adaptation to increased preload is a pathophysiological mechanism supported by many factors. According to Frank–Starling law, there is a positive relationship between preload and stroke volume. However, this relationship is not linear but rather traces a curve. Accordingly, once a concrete preload value has been reached, further increments do not give rise to significant additional stroke volume elevation. Preload dependency is the capacity of the heart to modify stroke volume in response to changes in preload, and depends on the basal preload value and the zone of the Frank–Starling curve in which both ventricles are operating. Each individual patient can present a series of ventricle function curves, and therefore a different response for the same preload increase. In haemodynamicly unstable patients, the management of fluid therapy is crucial because only 50% of patients are preload responders [13]. In our study, 13 (41%) patients had an increase in stroke volume after FC but no difference in clinical data were found to explain this different begaviour. Interindividual variations in body composition, basal circulating state, and coping with fasting may have influenced FC results. Future larger experimental studies are required to explore the different pathophysiological mechanisms involved in preload dependency of healthy volunteers.
In the limitation section, you could have added that these data were not compared with other methods, at least MRI, if not right cardiac catheterism (which is to understand because your subjects were healthy volunteers and the recordings were performed some time ago).
Response: We thank the reviewer for this remark. We agree with the reviewer but the availability of MRI and the difficult use of right heart catheter in healthy volunteers did not allowed us to made these comparisons. The duration of image acquisition and the availability of our CMR were the main reason. Besides, CMR and echocardiography techniques cannot be used interchangeably to determine and monitor RV strain. We added this remark in the limitation section, page 15, line 485, as follow = “The results of our study have not been compared with other techniques as CMR especially because CMR and echocardiography techniques are not interchangeable for RV strain evaluation[26].”
Please check again the grammar of the manuscript: there are several places where there are discrepancies between nouns and verbs.
Response: We thank the reviewer for this remark. We apologize for any typos errors and have corrected them.

Reviewer 2 Report
- The meaning of the "TAPSE" was not included at the beginning of the article and the abstract (it was specified only in line 41);
- The abbreviation "2D-STE TAD" appeared in line without being previously totally clarified;
- The authors wrote "Few data are available regarding their preload-dependency.", in line 17, and it is necessary to specify that it refers to the right ventricle.
- In line 18 "Methods: The acute modifications of RV preload was performed by a fluid challenge (FC)." it is necessary to be specific about the volume, amount of fluid, and velocity of infusion, as well as the position of the volunteers (in bed)? supine?)
- In line 19 the authors wrote, "Preload dependency (responder 19 group) was confirmed by an increase of at least 10% of the stroke volume measured by echocardiography after the FC.". With what basis of a similar other experiment was that methodology chosen? Also, references for this are missing;
- From lines 21 to 24 the presentation in numbers of the results are confused and the authors should follow the orientations for publications of number results available in the page of the JCM Journal;
- In line 25 and also line 28 appears "... 2D-STE parameters derived ..." but previously in this section appeared 2DE-STE-RV. The authors should make it uniformly presented;
- In line 30 the authors should also include preload increase with volume change in the "keywords" section;
- In line 41 the authors wrote "(TAPSE) or the systolic tricuspid annular velocity (RV-S’) [1]" but the correct should be and and instead of or";
- From line 45 to 48 "Two-dimensional speckle-tracking echocardiography (2D-STE) is a relatively new non- 45 invasive echocardiography technique that allows an objective and quantitative assessment 46 of the RV systolic function using simple parameters such as RV longitudinal strain or 47 tricuspid annular displacement (TAD) [1–3]. " the authors should analyze into more details the initial as well as the most important references focusing on the aims of such basic references of 2DE speckle tracking & RV analysis;
- In line 49 the authors discussed prognostic considerations, as "...but also their established 49 prognostic value (2,7)". This, the prognostic study, was not the aim of this experiment, and a statement-making it clear could be useful for the readers;
- In line 55 the limitation "...Current 55 guidelines do not provide references values for 2D-STE parameters, but rather lower limits 56 of normality for vendor-dependent software [8]." should clearly and well pointed;
- From lines 61 to 63 the authors wrote "... There are very few data regarding the direct influence of an acute change in preload on 2D-STE parameters in patients without the cardiopulmonary disease [10, 11]. Here comes such references. "10. Susilovic-Grabovac Z, Obad A, Duplančić D, et al (2018) 2D speckle tracking echocardiography of the right ventricle free wall 484 in SCUBA divers after single open sea dive. Clin Exp Pharmacol Physiol 45:234–240. https://doi.org/10.1111/1440-1681.12883 485 11. García-López ZY, Vargas-Barrón J (2019) Evaluation of the global longitudinal strain and segmental strain of the right ventricle 486 with two-dimensional speckle-tracking echocardiography with an elevation of the legs. ACME 89:2807. 487 https://doi.org/10.24875/ACME.M19000003". What called the attention is that the experiments published in these two references did not consider fluid challenge, one is with scuba diving and the other with an elevation of the legs. A reference more specific is missing;
- In line 68 appears "The aim of this study was to evaluate the effect of an acute change in preload on 2D-STE RV parameters in healthy volunteers. "This should be clarified bu being clear about the method that the authors proposed and accomplished in the study;
- In line 88 the authors wrote ",,,Protocol study 89 The volunteers’ blood volume was standardized as best as possible. All volunteers 90 had been fasting from 00:00 am and were included in the study between 8:00 and 11:00 91 a.m.". The point is that the volunteers could still have different basal circulating considering all variables that differ different body wrights ((more weight in terms of more adipose tissue) and gender. This can be a limitation of such study and if so has to be pointed in the adequate section of this article. ;
- Regarding the echocardiographic method in line 94 to line 135 "...The post-FC TTE was performed 15 minutes after the 94 FC.", the questions are: were the volunteers exactly in the same body positions? were the sonographer the same? were the preset of the equipment exactly the same? Was the echo machine the same one? were the records made exactly in the same way?
- From line 150 to 161 "... TAD analysis STE-based TAD measurements were also performed on the apical four-chamber. Three points were used for initialization on the first diastolic frame. These points were placed 1) at the insertion of the anterior tricuspid valve leaflet (RV free wall) and 2) of the septal leaflet into the tricuspid annulus, and 3) at the RV apex (Figure 2C). The software automatically tracked and calculated the TAD at the RV free wall (TADlat) and at the interventricular septum (TADsep). RV longitudinal shortening (RV-LS) was calculated as maximum end-systolic displacement (LES) of the mid-annular point from the measured annular motion and expressed in percent of the end diastolic RV longitudinal dimension (LED): 100 × (LED – LES)/LED (Figure 2C). The mid-annular point is automated selected by the software. All strain measurements were performed off line by the same observer (CB) blinded to fluid responsiveness status. 2D-STE parameters were analysed in one loop and results were the average of 3 measured. "... The authors should specify about the way they made their choice of points and offer references ideally of their own laboratory pointing on the appropriateness of the methodology;
- What was the reason to choose the statistical method of inter‐observer reliability, and intra-observer reliability by using intra-class correlation coefficient in order to do this analysis?
- In line 78 appears "32 healthy volunteers were included from September 2018 to November 2019. T..." Why such a long time of data acquisition? Did the authors took into consideration that this study was conducted in different weather conditions (and way of living of the volunteers?) from summer to winter (different habits? different indulgencies? different loading basal states? possibly different diets?)
- It called the attention that this is a very small number of volunteers, mostly males (*7%), and this is a serious limitation of such a study.
- There are patients in the series that modernly could be presenting themselves with arterial hypertension (over 120 mmHg systolic and/or 80 mmHg for diastolic measurements) and this can really be a limitation when analyzing the data of this study;
- To point in Table 2 line 246 only "Delta IVC (%)" statistically changed;
- in line 312 to 314 The authors began the discussion by writing".... Discussion 312 To our knowledge, this is the first study to investigate the preload dependency of 313 2D-STE RV parameters in young healthy volunteers." It should be specified here details about the fluid challenge;
- In line 326 when comparing both imaging cardiac methods it should be stressed that although 2-DE is more practical and more available there is a big difference in the CMR studies considering that they are always based in a 3-D visualization of cardiac chambers;
- In the lines of the section of discussion beginning in line 312 to line 408 there are many problems in the format of the writing (lack of comma, absence of separation of words, of symbols, and this all should be corrected, possibly should be done again because the meaning is sometimes confused;
- The reviewer stresses this limitation as pointed in line 420: "One another important limitation is that, unlike RVFWLS or RV4CSL which are widely available, TAD parameters are only provided by Philips Healthcare and not by other constructors." and also this study limitation in line 423: "...Recent studies have shown that 423 the three-dimensional approach can improve the assessment of global and regional RV systolic function [20, 21].";
- Finally, there is a limitation of being a single-center study.
Author Response
Reviewer 2
- The meaning of the "TAPSE" was not included at the beginning of the article and the abstract (it was specified only in line 41);
- The abbreviation "2D-STE TAD" appeared in line without being previously totally clarified;
Response: we thank the reviewer for this remark. We clarified the TAPSE and TAD parameters in the abstract section.
- The authors wrote "Few data are available regarding their preload-dependency.", in line 17, and it is necessary to specify that it refers to the right ventricle.
Response: we thank the reviewer for this remark. We clarified the sentence in the abstract section, page 1, line 16, as follow: Few data are available regarding 2D-STE parameters and their dependence on RV preload.
- In line 18 "Methods: The acute modifications of RV preload was performed by a fluid challenge (FC)." it is necessary to be specific about the volume, amount of fluid, and velocity of infusion, as well as the position of the volunteers (in bed)? supine?)
Response: we thank the reviewer for this remark. We complete the sentence page 1, line 19, as follow: Acute modification of RV preload was performed by a fluid challenge (FC): an infusion of 500 ml of 0.9% sodium chloride was given over 5 minutes in supine position. Preload dependency (responder group) was confirmed by a stroke volume increase of at least 10% measured by echocardiography after the FC.
- In line 19 the authors wrote, "Preload dependency (responder 19 group) was confirmed by an increase of at least 10% of the stroke volume measured by echocardiography after the FC.". With what basis of a similar other experiment was that methodology chosen? Also, references for this are missing;
Response : we thank the reviewer for this remark. We used the recommended echocardiographic method for measure the stroke volume (SV). Description of the measure was added in the method section, page 4, line 123 as follow: Doppler stroke volume (SV) was estimated by the product of LVOTarea and LVOT VTI [1].
We added the references concerning the definition of the preload dependence by the fluid challenge, page 3, line 94, as follow : The acute modification of the subjects’ RV preload was performed by a FC [12]. According to Frank-Starling relationship [13], preload dependency (responder group) was confirmed by an increase of at least 10% in stroke volume (SV) measured by echocardiography after the FC. Based on previous studies and on the variability of SV measurement, 10% increase is considered as clinically significant.
- From lines 21 to 24 the presentation in numbers of the results are confused and the authors should follow the orientations for publications of number results available in the page of the JCM Journal.
Response: We thank the reviewer for this remark. As suggested, we changed and clarified results in the abstract section according the JCM journal guidelines.
- In line 25 and also line 28 appears "... 2D-STE parameters derived ..." but previously in this section appeared 2DE-STE-RV. The authors should make it uniformly presented;
Response: we thank the reviewer for this remark. We changed the sentence and clarified this point in the manuscript.
- In line 30 the authors should also include preload increase with volume change in the "keywords" section;
Response: we thank the reviewer for this remark. We added the word preload and fluid challenge in the keywords section.
- In line 41 the authors wrote "(TAPSE) or the systolic tricuspid annular velocity (RV-S’) [1]" but the correct should be and and instead of or";
Response: we thank the reviewer for this remark. We corrected the error.
- From line 45 to 48 "Two-dimensional speckle-tracking echocardiography (2D-STE) is a relatively new non- 45 invasive echocardiography technique that allows an objective and quantitative assessment 46 of the RV systolic function using simple parameters such as RV longitudinal strain or 47 tricuspid annular displacement (TAD) [1–3]. " the authors should analyze into more details the initial as well as the most important references focusing on the aims of such basic references of 2DE speckle tracking & RV analysis;
Response: We thank the reviewer for this remark. We added a more specific reference and targeted references regarding the use of strain or TAD parameters for the analysis of right ventricular systolic function, page 2, line 47, as follow: Two-dimensional speckle-tracking echocardiography (2D-STE) is a relatively new non-invasive echocardiography technique that allows an objective and quantitative assessment of the RV systolic function [2] using simple parameters such as RV longitudinal strain [3] or tricuspid annular displacement (TAD) [4].
- In line 49 the authors discussed prognostic considerations, as "...but also their established 49 prognostic value (2,7)". This, the prognostic study, was not the aim of this experiment, and a statement-making it clear could be useful for the readers;
Response: We thank the reviewer for this remark. We change and clarify the sentence, page 2, line 53, as follow: [6] but also their established prognostic value (which is not the aim of the present experiment) (2,7)
In line 55 the limitation "...Current 55 guidelines do not provide references values for 2D-STE parameters, but rather lower limits 56 of normality for vendor-dependent software [8]." should clearly and well pointed;
Response: we thank the reviewer for this remark. In order to clarify the difficulties of measurement and interpretation of the 2D-STE parameters, we have added a reference, page 2, line 61, as follow: Recently, international recommendations have been made to standardize the measurement, interpretation and use of 2D-STE parameters [3].
- From lines 61 to 63 the authors wrote "... There are very few data regarding the direct influence of an acute change in preload on 2D-STE parameters in patients without the cardiopulmonary disease [10, 11]. Here comes such references. "10. Susilovic-Grabovac Z, Obad A, Duplančić D, et al (2018) 2D speckle tracking echocardiography of the right ventricle free wall 484 in SCUBA divers after single open sea dive. Clin Exp Pharmacol Physiol 45:234–240. https://doi.org/10.1111/1440-1681.12883 485 11. García-López ZY, Vargas-Barrón J (2019) Evaluation of the global longitudinal strain and segmental strain of the right ventricle 486 with two-dimensional speckle-tracking echocardiography with an elevation of the legs. ACME 89:2807. 487 https://doi.org/10.24875/ACME.M19000003". What called the attention is that the experiments published in these two references did not consider fluid challenge, one is with scuba diving and the other with an elevation of the legs. A reference more specific is missing;
Response; We thank the reviewer for this remark. There are no data regarding the direct influence of an acute change in preload by FC on 2D-STE parameters in patients without cardiopulmonary disease hence the interest of our study. Previous experimental studies used non-invasive techniques to increase RV preload such as the passive leg rising [11, 12].
We have completed the sentence and modified references, page 2, line 67, as follow: There are no data regarding the direct influence of an acute change in preload induced by FC on 2D-STE parameters in patients without cardiopulmonary disease. Previous experimental studies used surrogates of FC, as passive leg rising (that mobilizes blood from the venous compartment) to increase RV preload [11, 12].
- In line 68 appears "The aim of this study was to evaluate the effect of an acute change in preload on 2D-STE RV parameters in healthy volunteers. "This should be clarified bu being clear about the method that the authors proposed and accomplished in the study;
Response: we thank the reviwer for this remark. We clarify the sentence, page 2, line 77, as follow: The aim of this study was to evaluate the effect of an acute change in preload on 2D-STE parameters in healthy volunteers. This acute change in preload was performed by an infusion of 500 ml of saline over 5 minutes. The preload dependency of 2D-STE parameters was defined by a 10% increase of stroke volume (SV) measured by transthoracic echocardiography after FC.
- In line 88 the authors wrote ",,,Protocol study 89 The volunteers’ blood volume was standardized as best as possible. All volunteers 90 had been fasting from 00:00 am and were included in the study between 8:00 and 11:00 91 a.m.". The point is that the volunteers could still have different basal circulating considering all variables that differ different body wrights ((more weight in terms of more adipose tissue) and gender. This can be a limitation of such study and if so has to be pointed in the adequate section of this article. ;
Response: We thank the reviewer for this comment. We complete and clarify this limitation in the limitation section, page 15, line 468, as follow: ”Moreover, different shapes, body compositions and basal circulating states of the subjects may have influenced FC results. For further studies, a Bioelectrical impedance analysis, for example, may allow a more accurate assessment of basal state and fluid distribution [23]”.
- Regarding the echocardiographic method in line 94 to line 135 "...The post-FC TTE was performed 15 minutes after the 94 FC.", the questions are: were the volunteers exactly in the same body positions? were the sonographer the same? were the preset of the equipment exactly the same? Was the echo machine the same one? were the records made exactly in the same way?
Response: we thank the reviewer for this remark. The TTE protocol was standardized, performed in supine position, with a high-quality commercially available ultrasound system (CX 50, Philips Healthcare) and by a single experienced cardiologist. We completed the method section, page 4, line 116, as follow: A standardized TTE protocol was performed according to recent guidelines [1] and using high-quality commercially available ultrasound system (CX 50, Philips Healthcare). The standardized TTE protocol was performed in supine position by a single experienced cardiologist (CB)
- From line 150 to 161 "... TAD analysis STE-based TAD measurements were also performed on the apical four-chamber. Three points were used for initialization on the first diastolic frame. These points were placed 1) at the insertion of the anterior tricuspid valve leaflet (RV free wall) and 2) of the septal leaflet into the tricuspid annulus, and 3) at the RV apex (Figure 2C). The software automatically tracked and calculated the TAD at the RV free wall (TADlat) and at the interventricular septum (TADsep). RV longitudinal shortening (RV-LS) was calculated as maximum end-systolic displacement (LES) of the mid-annular point from the measured annular motion and expressed in percent of the end diastolic RV longitudinal dimension (LED): 100 × (LED – LES)/LED (Figure 2C). The mid-annular point is automated selected by the software. All strain measurements were performed off line by the same observer (CB) blinded to fluid responsiveness status. 2D-STE parameters were analysed in one loop and results were the average of 3 measured. "...
The authors should specify about the way they made their choice of points and offer references ideally of their own laboratory pointing on the appropriateness of the methodology
Response: We thank the reviewer for this comment. We used the classical and described echocardiographic method for assessing TAD parameters. We clarify this point and added specific references, page 5, line 180, as follow: The echocardiographic method used to measure TAD parameters was identical to that described in previous studies [4, 13].
- What was the reason to choose the statistical method of inter‐observer reliability, and intra-observer reliability by using intra-class correlation coefficient in order to do this analysis?
Response: We thank the reviewer for this remark. The Intra-class correlation coefficient is the recommended test for assessing the inter and intra observer reliability. We added the specific reference, page 6, line 190, as follow: The inter‐observer and intra‐observer's reliability of 2D-STE measurements were assessed using intraclass correlation coefficient (ICC) as recommended [14].
- In line 78 appears "32 healthy volunteers were included from September 2018 to November 2019. T..." Why such a long time of data acquisition? Did the authors took into consideration that this study was conducted in different weather conditions (and way of living of the volunteers?) from summer to winter (different habits? different indulgencies? different loading basal states? possibly different diets?)
Response: We thank the reviewer for this remark. This is an ancillary study of the PORTEAU study evaluating portal flow in MRI. The low availability of MRI slots dedicated to clinical research explains the duration of the study. The ultrasound protocol was standardized, performed in the same room and under the same ambient temperature conditions throughout the study. Eating habits or diets were not assessed. A minimum of 8-hour fasting was performed for all participants.
- It called the attention that this is a very small number of volunteers, mostly males (*7%), and this is a serious limitation of such a study.
Response: We are aware of the small size of our study. To date, this is the only study evaluating the effect of FC on 2D-STE parameters. Therefore, this limitation is described in the limitation section, page 15, line 485, as follow: Finally, this study involved a limited number of young volunteers, especially males, in a single-centre. Further larger studies are required to confirm our findings.
- There are patients in the series that modernly could be presenting themselves with arterial hypertension (over 120 mmHg systolic and/or 80 mmHg for diastolic measurements) and this can really be a limitation when analyzing the data of this study;
Response: we thank the reviewer for this remark. All the subjects' blood pressure measurements (see table 1, SAP (119 [115-128] mmHg and DAP 68 [64-77] mmHg before FC) were below the French and ESC recommendations for the diagnosis of hypertension (https://doi.org/10.1093/eurheartj/ehy686) The age and low body weight of the patients are not in favor of secondary hypertension.
- To point in Table 2 line 246 only "Delta IVC (%)" statistically changed;
Response: We thank the reviewer for this remark. We added the result of the IVC collapsibility index, page 8, line 247, as follow: In the responder group, IVC collapsibility index decreased after FC (from 35.5 [28-55] % to 24 [22-30], p=0.001).
- in line 312 to 314 The authors began the discussion by writing".... Discussion 312 To our knowledge, this is the first study to investigate the preload dependency of 313 2D-STE RV parameters in young healthy volunteers." It should be specified here details about the fluid challenge;
Response: we thank the reviewer for this remark. We clarify the sentence about the use of FC, page 12, line 343, as follow: To our knowledge, this is the first study to investigate the RV preload dependency of 2D-STE parameters assessed by performing a FC of 500ml of 0.9% sodium chloride, in young healthy volunteers.
- In line 326 when comparing both imaging cardiac methods it should be stressed that although 2-DE is more practical and more available there is a big difference in the CMR studies considering that they are always based in a 3-D visualization of cardiac chambers;
Response: We thank the reviewer for this remark. We complete the discussion section with this remark, page 12, line 359, as follow: The introduction of new echocardiographic parameters, such as 2D-STE parameters, has significantly improved our ability to assess RV systolic function. These parameters are highly correlated with CMR measurements even if CMR evaluate RV in 3 dimensions [13]. However, 2D-STE parameters are more available and feasible at the patient’s bed side
- In the lines of the section of discussion beginning in line 312 to line 408 there are many problems in the format of the writing (lack of comma, absence of separation of words, of symbols, and this all should be corrected, possibly should be done again because the meaning is sometimes confused;
Response: We thank the reviewer for this remark. As suggested, we have tried to modify and improve the reading of the discussion.
- The reviewer stresses this limitation as pointed in line 420: "One another important limitation is that, unlike RVFWLS or RV4CSL which are widely available, TAD parameters are only provided by Philips Healthcare and not by other constructors." and also this study limitation in line 423: "...Recent studies have shown that 423 the three-dimensional approach can improve the assessment of global and regional RV systolic function [20, 21].";
Response: We thank the reviewer for this remark. We agree that the need to use Philips software limits the use of TAD parameters. That's why we thought it was important to include it in the limitation section. Philipps is a world leader in the field of echocardiography and the development of the parameters is only at its beginning.
- Finally, there is a limitation of being a single-center study.
Response: We thank the reviewer for this remark. We added this limitation, in the limitation section, page 15, line 486, as follow: Finally, this study involved a limited number of healthy young volunteers, mostly males, in a single-center”.

Reviewer 3 Report
In their paper, the Authors focused on a very actual and significant topic such as the dependency from preload conditions in the assessment of right ventricle (RV) systolic function using 2D speckle tracking echocardiography (STE).
RV-STE is emerging as a strong prognostic marker in many clinical settings such as heart failure with reduced ejection fraction (J Am Coll Cardiol Img. 2019 Dec, 12 (12) 2373–2385), tricuspid regurgitation (European Heart Journal, Volume 42, Issue Supplement_1, October 2021, ehab724.0117) or pulmonary hypertension (Can J Cardiol. 2018 Aug;34(8):1069-1078).
Strain imaging techniques have emerged as a very attractive, feasible and valuable clinical tool for the analysis of myocardial performance; however, the appropriate interpretation of measurements requires an understanding of cardiac mechanics and many factors potentially influencing the measurements such as, for example, loading conditions, must be taken into account, especially in pathological scenarios.
Given these premises, the Authors evaluated and discusses the effects of an acute change in preload in 2D-STE RV parameters in healthy volunteers.
The paper is well written and extremely clear and precise.
I believe that the following points should be addressed.
- Abstract: The acronym “TAPSE” should be explained.
- Line 326: The reference n. 13 should be put into brackets.
Author Response
Reviewer 3
In their paper, the Authors focused on a very actual and significant topic such as the dependency from preload conditions in the assessment of right ventricle (RV) systolic function using 2D speckle tracking echocardiography (STE).
RV-STE is emerging as a strong prognostic marker in many clinical settings such as heart failure with reduced ejection fraction (J Am Coll Cardiol Img. 2019 Dec, 12 (12) 2373–2385), tricuspid regurgitation (European Heart Journal, Volume 42, Issue Supplement_1, October 2021, ehab724.0117) or pulmonary hypertension (Can J Cardiol. 2018 Aug;34(8):1069-1078).
Strain imaging techniques have emerged as a very attractive, feasible and valuable clinical tool for the analysis of myocardial performance; however, the appropriate interpretation of measurements requires an understanding of cardiac mechanics and many factors potentially influencing the measurements such as, for example, loading conditions, must be taken into account, especially in pathological scenarios.
Given these premises, the Authors evaluated and discusses the effects of an acute change in preload in 2D-STE RV parameters in healthy volunteers.
The paper is well written and extremely clear and precise.
I believe that the following points should be addressed.
- Abstract: The acronym “TAPSE” should be explained.
- Line 326: The reference n. 13 should be put into brackets.
We thank the reviewer for his comments and for his interest in our study. We corrected the aforementioned points.

Round 2
Reviewer 2 Report
I acknowledge the author's response and I recognize that with these points that they addressed the manuscript could eventually be useful for readers. The future of investigations in the field is wide open and the eyes and the ears of the specialized community are willing to see and to hear about research in this particular field. The best way to use the principles of stress echo in right ventricle research necessarily goes through continuous manufacturers subside and there is a hope that it will be available in the future in other manufacturers as well to pursue in the field.